# Nanoparticles for Coronavirus Control

**DOI:** 10.3390/nano12091602

**Published:** 2022-05-09

**Authors:** Maryam Kianpour, Mohsen Akbarian, Vladimir N. Uversky

**Affiliations:** 1Institute of Biomedical Sciences, National Sun Yat-sen University, Kaohsiung 804, Taiwan; maryam.kianpour89@gmail.com; 2Department of Chemistry, National Cheng Kung University, Tainan 701, Taiwan; 3Department of Molecular Medicine and Health Byrd Alzheimer’s Institute, Morsani College of Medicine, University of South Florida, Tampa, FL 33612, USA; 4Laboratory of New Methods in Biology, Institute for Biological Instrumentation of the Russian Academy of Sciences, Federal Research Center ‘‘Pushchino Scientific Center for Biological Research of the Russian Academy of Sciences’’, 142290 Pushchino, Moscow Region, Russia

**Keywords:** coronaviruses, vaccines, nanoparticles, diagnostic nanoparticles

## Abstract

More than 2 years have passed since the SARS-CoV-2 outbreak began, and many challenges that existed at the beginning of this pandemic have been solved. Some countries have been able to overcome this global challenge by relying on vaccines against the virus, and vaccination has begun in many countries. Many of the proposed vaccines have nanoparticles as carriers, and there are different nano-based diagnostic approaches for rapid detection of the virus. In this review article, we briefly examine the biology of SARS-CoV-2, including the structure of the virus and what makes it pathogenic, as well as describe biotechnological methods of vaccine production, and types of the available and published nano-based ideas for overcoming the virus pandemic. Among these issues, various physical and chemical properties of nanoparticles are discussed to evaluate the optimal conditions for the production of the nano-mediated vaccines. At the end, challenges facing the international community and biotechnological answers for future viral attacks are reviewed.

## 1. Introduction

Human life has always been in danger of various sources of harm. Millennia ago, our ancestors thought that all of their problems were limited to finding enough food and staying safe from predators. Later, with the advancement of science, more civilized humans were able to partially overcome these long-standing challenges but were unaware that new problems would arise [1]. During the entire history of humankind, we have used our knowledge to raise the quality of life. However, the mortality rates caused by numerous maladies, such as neurodegeneration, contagions (including various viral infections), cardiovascular disorders, diabetes, and different forms of cancer remain high. Although some of the current health issues, such as cancer, diabetes, cardiovascular diseases, and neurodegeneration, are associated with the modern lifestyle and can potentially be reduced by changing habits, exposure to dangerous microorganisms and viruses clearly cannot be easily ignored. Furthermore, with its ever-increasing connectivity among countries and continents, the modern world is clearly facing global risks of fast spread of dangerous infections. These days, many countries in the world are struggling with the second and third waves of the coronavirus (CoV) disease 2019 (COVID-19) caused by severe acute respiratory syndrome coronavirus-2 (SARS-CoV-2) infection. Every day, thousands of people around the world fall victim to this virus. Due to COVID-19, the economies of countries are worsening daily, many businesses and industries have gone bankrupt, and millions of people have lost their jobs [2,3,4,5]. In this sense, the global SARS-CoV-2 pandemic filled the 21st century with serious challenges.

In the last two decades, the human race has been repeatedly attacked by human coronaviruses (HCoVs). In 2002, the severe acute respiratory syndrome CoV (SARS-CoV) claimed 8096 lives, whereas 2494 people succumbed to the Middle East respiratory syndrome CoV (MERS-CoV) in 2012, and finally the uninvited guest of 2019–2022, SARS-CoV-2, is still collecting its toll. In comparison with the rather moderate incidences of SARS-CoV and MERS-CoV, the occurrence of the SARS-CoV-2 is significantly higher [6,7,8,9]. As of June 2021, 176,482,998 people were reported to be infected with SARS-CoV-2, of which 3,812,194 died [8]. According to the Worldometer (https://www.worldometers.info/coronavirus/), as of 23 April 2022, the number of SARS-CoV-2 infected individuals climbed to 509,048,132, and 6,241,704 COVID-19 patients passed away.

HCoVs, which were discovered in 1960, belong to the orthocoronavirinae subfamily belonging to the coronaviridae family. There are seven HCoVs, which, in addition to the aforementioned SARS-CoV, SARS-CoV-2, and MERS-CoV, include HCoV-229E, HCoV-NL63, HCoV-OC43, and HCoV-HKU1 types that usually cause mild-to-moderate upper-respiratory tract illnesses, with symptoms similar to the common cold. HCoV-229E and HCoV-NL63 are alpha CoVs, whereas SARS-CoV, SARS-CoV-2, MERS-CoV, and the lesser-known types of HCoV-OC43 and HCoV- HKU1 all belong to the beta CoV genus [10]. This genus in particular has a high potential for infecting humans. All CoVs are capsid-coated viruses, usually spherical, and contain a single-stranded RNA genome with a length of 32–37 kb [11,12,13].

With a deeper look at health problems and by studying modern ways to deal with them, we can boldly say that the obvious human hope to combat most of these diseases is the use of proteins and vaccines [14,15,16]. In the last three decades, with the advent of biotechnology and nanotechnology for the recombinant production of human proteins in different hosts, diverse medicinal proteins have found their way into the field of treatment for the management of many diseases. To date, nearly 400 types of these biologics have been introduced [17,18]. These proteins could potentially reach the market faster than chemically synthesized drugs due to better predictability of their behavior in the living environment, as well as their lower toxicity [19,20]. In this study, with the help of recent data, we first describe the structure of SARS-CoV-2. In the next section, we look at how the virus causes pathogenesis and the body’s immune response to this infection. Useful information on the use of nanoparticles in the production of the coronavirus vaccines such as DNA vaccine, RNA vaccine, viral vector vaccine, and adenovirus-vector vaccine will also be provided. The use of monoclonal antibodies in passive immunization/adaptive immunity and the 2022 status of the virus vaccines will be summarized. Finally, the challenges in the field of vaccination, problems related to vaccines, and solutions to overcome them will be presented in the form of a list of suggestions. Also this paper will describe the utilization of various types of polymer nanoparticles, metal nanoparticles, and peptide nanoparticles for the detection and suppression of coronavirus infection. Properties of these nanoparticles, such as particle size, surface charge, particle shape, and hydrophobicity and hydrophilicity will also be examined in the types of immunological responses nanoparticles may generate. The use of nanoparticles in SARS-CoV-2 diagnostic methods and the strategies used will also be reviewed.

## 2. SARS-CoV-2 Structure and Infection Mechanism

Knowing the structural features and life cycle peculiarities of the virus will enable researchers to suggest solutions to deal with the virus outbreak. With their genomes approaching 30 kb in length, CoVs are among the largest known RNA viruses. SARS-CoV-2 is an enveloped positive-strand single-strand RNA virus (+ssRNA virus), whose genomic ssRNA is condensed by the nucleocapsid (N) proteins at the center of the viral particle. The size of the viral particle of SARS-CoV-2 can be up to 100 nm [21]. The outermost layer of the viral particle is made of a phospholipid membrane similar to mammalian cells, which contains three types of viral proteins. These proteins include membrane (M) protein in high abundance, coating proteins (envelop protein, E protein) in relatively low abundance, and most importantly, spike protein (S protein) [22,23]. S protein is a trimeric glycoprotein, a monomer that has a total length of 1273 amino acids. There is a 76% sequence similarity between the S proteins of SARS-CoV and SARS-CoV-2, and they are glycosylated at 21 to 35 sites, respectively.

The S protein consists of two main parts called S1 and S2 subunits. The S1 subunit has a segment that detects mammalian cellular receptors (receptor-binding domain, RBD) and is responsible for binding the viral particle to the host cell, while the S2 subunit has trans-membrane domains that hold the protein like a rod into the viral membrane [21,24]. Thanks to the structural studies of this protein, it was observed that as soon as S protein binds to receptors on the surface of the host cell, the open structure of this protein becomes closed. This structural switch ensures a strong attachment between the virus and the host cell.

M and E proteins play structural roles for viral particles [25,26,27]. Figure 1 illustrates the overall view of a CoV particle. Attachment to the surface of the host cell followed by cell entry is among the most important aspects of the life cycle [28]. A correct and in-depth understanding of this stage of viral infection can help in designing drugs to prevent the virus entrance into the host cell. Although this stage of the virus’ life cycle has not yet been completely uncovered, significant progress has been made in this field. In this article, a detailed description of the stages of viral infection will be presented, and the mechanisms that have been proposed to fight the virus at each step will be discussed.

To design a safe drug, one has to understand the effects of the target virus on the body. Figure 2 presents details of how the body responds to the virus and shows that the virus first enters the lungs through the respiratory system, and then the immune system in the alveoli would be used as the first level of defense against the virus.

To that end, immune cells that have entered through blood vessels into the alveoli, secrete a variety of cytokines in contact with the invading virus [30]. This response, by itself, triggers more immune cells that would generate more cytokines, resulting in a cytokine storm [31]. Among the most important cytokines secreted at this stage are the interleukins 6 and 1 (IL-6 and IL-1), as well as α-interferon (INF-α). The total secretion of cytokines causes the overproduction and exudation of fibrin in the alveoli, which, by disruption of cell junction, can eventually increase the flow of blood fluid through the capillaries to the lung chamber, resulting in pulmonary edema, focal hemorrhage, and pulmonary consolidation [32,33,34,35]. All of this occurs due to the body’s intense inflammatory responses against the invading virus [36].

Upon closer inspection of the infected cell, the first step in the virus entering the host cell is the interaction between the S protein (RBD of the S1 subunit) and the angiotensin-converting enzyme 2 (ACE2) receptor at the surface of the host cell [37,38,39]. The analog for this receptor in the MERS-CoV infection mechanism is dipeptidyl peptidase-4 (DPP4) receptor [40,41]. The angiotensin converting-enzyme is an important membrane protein that is abundantly expressed on the surface of cells of various human and animal tissues. The tissues with the highest expression levels of ACE2 are the lungs, gastrointestinal tract, blood vessels, kidneys, liver, and heart. The history of recognizing the importance of this protein dates back to three decades ago [42].

During these years, scientists speculated on many functions for this protein, including its crucial role in the renin–angiotensin–aldosterone system (RAAS) pathway regulating blood pressure, wound healing, and inflammation. Here, ACE2 modulates activities of angiotensin II (ANG II), a protein that increases blood pressure, body water and sodium content, and inflammation, as well as increases damage to blood vessel linings and various types of tissue injury, by converting ANG II to other molecules that counteract the effects of ANG II. Concerning the molecular mechanisms of CoV-2 infection, ACE2 serves as a major receptor controlling the main route of SARS-CoV-2 entry to the host cells [43].

The importance of this receptor is further emphasized by finding a close relationship between COVID-19 vulnerability and gastrointestinal symptoms due to high expression levels of the ACE2 receptor at the gastrointestinal epithelial cells [44,45]. However, some researchers have suggested that SARS-CoV-2 may also enter the host cell through interaction with another receptor, the CD147 protein [46], although this claim has been questioned in more recent studies [47]. This receptor is also expressed on the surface of many tissues and is responsible for changing the shape of the matrix in some phenomena, such as cancer, inflammation, and wound healing [48]. It is believed that the reason for the different responses of different patients to viral infection is a difference in the expression level of these receptors at the surface of their cells. With all this, as soon as the interaction between ACE2 and S protein has taken place, the viral particle enters the host cell by endocytosis, and the virus genome is released [49]. However, one should take a closer look at the roles of another protein in this process. The binding of S protein to ACE2 is not sufficient for cell entry, and the internalization of the virus particle to the host cell is activated by the specific cleavage of the ACE2-bound S protein by a transmembrane serine protease protein-2 (TMPSS2) [50,51]. As another serendipity of the human body, this protein has been closely linked to many health problems. For example, the crucial roles of TMPSS2 in the metastasis and progression of prostate cancer are well-known [52].

After entering the cell, due to the suitable conditions for viral protein translation, the host translational machinery starts producing viral RNA-dependent RNA polymerase, and this event marks the beginning of a cascade of intracellular processes in favor of the invading virus. After the viral polymerase is translated, a variety of viral RNA genes is produced, resulting in the expression of structural and functional proteins needed to form a complete viral particle inside the host cell [53,54]. Consequently, at the last stage of viral infection, whole viral particles are expelled by the explosion of the host cell or exocytosis, infecting the surrounding healthy cells. All of these pathways have been used to design strategies to combat the SARS-CoV-2 infection [55,56,57]. Although no promising and definitive approaches to combat CoVs infection and COVID-9 progression have been reported so far, in this article, we intend to introduce all the proposed strategies based on using the biological materials, or the combination of these materials with non-biological materials, to limit virus progression.

## 3. Our New Comrades-in-Arms: Nanomaterials and the Development of CoV-2 Vaccines

Nanomaterials have been used in various fields for decades. This group of materials in general, with their very large surface-to-volume ratios, the ability to place molecules with different properties on their surface (functionalization), and simplicity in their production process have played a major role in advancing human knowledge and increasing quality of life [58,59,60]. Nonetheless, although some scientists have discovered new usages for the substances, others pointed to their side effects. Thereby, this type of material should be considered a double-edged sword. In this section, different applications of nanomaterials in the inhibition of the virus life cycle are presented.

### 3.1. Different Classes of Nanomaterials against Coronavirus Disease

This ongoing pandemic, with all its bitterness, proved that the harmony between the sciences could be a great help in achieving human goals. For example, it was observed that the use of various nanoparticles along with molecular biology eventually introduced vaccines into the consumer markets, which turned the dark days of the epidemic into a beacon of hope. Scientists in the nanotechnology field have followed various strategies, each with a vision to generate means to fight the virus. Some nanoparticles have been used to directly fight the virus [61], while others have been used for the rapid detection of viruses in laboratory samples [62]. Additionally, some nanocarriers are used to deliver anti-SARS-Cov-2 drugs/vaccines [63]. In the subsequent sections of this study, the types of these nanoparticles will be discussed. In general, these nanoparticles are classified into groups based on their chemical nature, such as polymer nanoparticles, metal nanoparticles, and peptide nanoparticles. Each of these groups can be subdivided into more detailed subgroups on their own, which will be explained further.

To deal with global epidemics, careful review and utilization of all available tools/means are important. In this regard, the use of nanotechnology as a new field in medical sciences and its multifunctional structures can be a solution. Nanotechnology can be used for a variety of medical purposes, such as clinical diagnosis, pharmaceutical research, immune system activation, and the extraction of the biological materials. To defeat COVID-19, better understanding of the virus, better diagnosis of infection, its treatment and prevention are steps in which nanotechnology is expected to help [64].

Polymer nanoparticles have found their place in many industrial and medical fields. These substances, especially in regenerative medicine, have been able to give much hope to the patient community to recover from tissue degeneration diseases [65]. Having a high level of safety, biodegradability, simplicity of synthesis, and the ability to control their properties through different functionalizations initiated strong attention to nanoparticles in the field of anti-SARS-CoV-2 research [61]. Another positive point for this group of nanoparticles is that some of them have been approved by the Food and Drug Administration (FDA) and have been studied in great detail in other fields [66]. By selecting this group of nanoparticles, the distance to reach the final goal may not be as long as the one needed for the development of novel and completely unknown nanoparticles.

Certainly, the synthesis methods of these nanoparticles, which directly affect their properties, will also indirectly determine the effectiveness of nanoparticles in combating the new coronavirus. The sizes of nanoparticles, which are generally between 1 and 100 nanometers, guarantee a very high surface to volume ratio, as well as the ability to load significant amounts of drugs in small amounts of nanoparticles, which can help a lot in combating pathogens, including SARS-CoV-2 virus [67].

### 3.2. Antiviral Mechanism of Nanoparticles

The antiviral mechanisms of nanoparticles include inhibiting the binding of the virus to the target cell, preventing the virus from entering into the host cell, and attacking the growth and proliferation stage of the virus. Possible mechanisms of nanoparticles include direct and indirect inactivation of viruses. These mechanisms vary depending on the three-dimensional shape and type of nanoparticles [68,69]. Another mechanism proposed for the antiviral action of nanoparticles is the local field action of nanoparticles. In this way, the designed nanoparticles change the membrane potential at the surface of the host cell as soon as they are adsorbed on the cell surface. Following this, membrane potential change and the penetration of the virus into the host cell are affected and reduced [70].

Other studies have suggested that metal nanoparticles, such as those containing silver ions with oxidizing properties in infected host cells, can prevent the virus from spreading to the healthy cells [71]. According to an in silico study, iron nanoparticles have also been shown to form a stable complex with the CoV spike protein and prevent the virus from attaching to the host cell [72].

Table 1 lists some of the studies that used nanoparticles in fighting respiratory viral diseases [73].

### 3.3. Properties of Nanoparticles for Efficient Vaccine Production

Vaccines have shown great potential for use in the prevention and treatment of infectious diseases. With the rapid development of biotechnology and materials science, nanomaterials have found an essential place in the formulation of new vaccines, as they can enhance the effect of antigens by acting as a release system and/or as an immune-boosting aid [74]. The analysis of the effects of nanoparticles on vaccine properties shows the improvement of the antigen stability and immunogenicity as well as a capability of targeted and controlled release of active substances [74]. However, there are still obstacles in this field due to the lack of fundamental knowledge on how nanoparticles act at the molecular scale, and what the biological effects of nanoparticles in living organisms are [75]. Nanoparticle-based vaccines are classified based on the function of the nanoparticles in them as a release system and immune response enhancer. Therefore, a fundamental understanding of the distribution of nanoparticles in the body and their fate will accelerate the logical design of new nanoparticles that will change the future of vaccines. Nanotechnology has provided the opportunity to design different nanoparticles in terms of composition, size, shape, and surface properties for various pharmaceutical applications [76].

Nanoparticles with the same size as cellular components can show biophysical function and biotherapy similar to their biological counterparts. There are several systematic studies which showed that the nanoparticles designed with polyethylene glycol (PEG) are able to delay the clearance of the drug from the body and thus make the systematic circulation of the drug in the body longer than in the free drug state [77]. This can eventually be useful for the accumulation of more drug at the site of treatment. In addition, nanoparticle delivery systems can have several salient features, including high drug loading capacity, controlled release rate, and reduced drug toxicity in the body [78]. As a result, nanoparticle-based approaches as release systems provide new opportunities to enhance innate immune activation and induce a strong immune response to the slightest toxicity [75]. The most important components of an effective vaccine include an antigen to activate the immune system, an enhancer of the immune response to stimulate the innate immune system, and a release system to ensure proper antigen delivery and targeting [79]. To achieve these goals, the design of nanoparticles focuses on the chemical composition, size, surface charge, and surface properties of the nanoparticles, as these are used to control the distribution of these particles in the environment, the release of antigen, the efficiency of immune stimulation, and the final immune response [80]. Emulsions, liposomes, and synthetic polymers are nanoparticles that serve as helpers for the proper release of immune response enhancers. Antigen-carrying nanoparticles are able to affect the immune response and significantly enhance the T-cell cytotoxic response against the antigen fused to the nanoparticles [81]. This is due to the specialized ability of some antigen-presenting cells (APCs), which can effectively absorb foreign particles, such as microparticles and bacteria [82]. This process is performed by detecting antigenic material to analyze and express foreign antigens to other cells in the immune system. However, there are limitations to the utilization of these approaches, such as the existence of the nonspecific uptake and immunosuppression activities of these compounds.

As mentioned, controlling the size, shape, and chemical properties of nanoparticles enables these tiny particles to have a controllable cell uptake coefficient. In order to provide organized information for efficient vaccine production, Figure 3 and the discussion below indicate the properties that nanoparticles must have to be considered [80].

As shown in Figure 3, there are many criteria to consider when designing a nanoparticle-based vaccine. Nanoparticles with the size of less than ten nanometers are usually reported to be excreted by the kidneys, while larger particles are excreted by the liver [83]. Particles larger than 200 nm have also been shown to be able to be repelled by the spleen as long as they have good flexibility [84]. Therefore, in addition to the items listed in Figure 3, the flexibility of the nanomaterials is another important factor [85]. It has been seen that the shape of nanoparticles can also affect their cellular uptake and ultimately affect immune response they trigger [86]. Traditionally, the nanoparticles for formulations were considered spherical, but currently, different types of nanoparticles, such as disks, rods, prisms, and stars are being designed and studied [87]. It was reported that even the symmetry of nanomaterials is important for effective tissue distribution and cellular uptake, which can be due to the amount of reactions that occur at different levels [88].

It is also possible to generate positive and negative charges with different densities through chemical modifications applied to the surface of the nanoparticles. By applying this feature, the interaction of materials with targets is driven by electrostatic forces [80]. In many studies, the effects of different charges on immune responses were investigated. For example, positively charged hydrogel nanoparticles (modified with antigens) have been shown to stimulate antibody production, T-cell activation, and class II MHC expression [89]. The same effect was observed with hydrophobicity in mesoporous silica nanoparticles that affect the expression of CD3, 4 and 8 [90]. In terms of tissue penetration, it has also been observed that positively charged nanoparticles are capable of penetrating the skin 2–4 times more efficiently than their negatively charged counterparts [91].

In addition to the features reviewed in previous sections, the ligand density at the surface of nanoparticles also has a significant effect on the immunological response they generate. These differences in response may be due to the differences in cellular uptake. For example, it has been shown that the amino content of silica mesoporous particles significantly reduces the cytotoxicity of these particles, while PEGylation is effective in increasing the hydrophilicity degree and ultimately leads to an increase in the renal filtration rate [92,93].

### 3.4. Different Nanoparticle-Based Vaccines for CoVs

We focus here on the classification of nanoparticles that have been designed to fight CoVs. From a structural viewpoint, four types of nanoparticles have been introduced for this purpose (Figure 4). The first group of these nanoparticles originates from proteins that can self-assemble; e.g., viral proteins that can aggregate into virus-like particles or form protein micelles [94,95,96]. Similar to a virus particle, there is another group of nanoparticles that are liposomes along with the capsid proteins. Liposomes themselves can also be considered as nanoparticles capable of carrying therapeutic agents inside their body against CoV. Finally, exosomes are another group of nanoparticles that are very similar to viruses, except that these particles are usually produced by exocytosis from virus-infected cells [97,98].

Today, perhaps the most important application of nanoparticles due to the global CoV-2 challenge is the use of these materials to load and transport viral antigens and viral DNA or RNA genomes [99,100]. In the meantime, physical and chemical interactions, such as adsorption, entrapment, and attachment have been used to load viral materials into the nanoparticles. For this important purpose, a variety of nanoparticles, such as nanopolymers [101], liposomes [102,103], and quantum dots [104,105] have been used [100]. Between the years 2014–2018, scientists used protein micelles consisting of the S protein of SARS-CoV-1 and MERS-CoV to fight these viruses [106,107]. Of these, some remain in the early clinical stages, but recently, this method has also been used to deal with the SARS-CoV-2 infection.

The vaccine-related virus-mimicking nanoparticles (NPs) such as self-assembled viral proteins and virus-like particles are in phase I clinical trials [97]. The advantages of using this group of nanoparticles include simplicity of their production, safety, high resistance in vitro, and virus-like body distribution. However, this type of nanoparticle also has some disadvantages, such as high production cost, the difficulty of industrialization, low stability in vivo, and occurrence of unwanted immune reactions [97]. Studies to suppress the progression of MERS-CoV [108,109] and SARS-CoV-2 [97] have been performed using virus-like particles of S proteins and RBD domains, respectively. Both of these approaches are in the early clinical phase.

Virus-like particles have very high immunogenicity and have recently been considered for their various applications in vaccination, targeted drug delivery, gene therapy, and immunotherapy. All four recombinant vaccines—Engerix, Cervarix, Recombivax HB, and Gardasil—on the market are based on highly pure virus-like particles (VLPs) [110]. However, there are several potential barriers in the development of virus-based vaccines from the research phase to the clinical phase. One of these problems is the lack of information on the folding and proper structure of these particles as compared with the parent infectious virus. Another problem with these particles is that the binding pattern of these particles is not the same as that of the parent infectious virus. Although these particles contain capsid proteins and can stimulate the body’s immune system, they lack other viral components. Another problem is the complexity of the related clinical studies. However, human health has historically been more valuable than the problems and shortcomings that are considered for these vaccines [111]. Virus-like particles can be divided into two main categories based on the structure of parental viruses [112]: non-enveloped viral particles and enveloped viral particles. Non-enveloped virus-like particles typically consist of one or more pathogenic components that are self-assembled into the particles. These particles also do not contain any of the host components. This kind of VLP has been used to develop vaccines against pathogens such as HPV and RV [112]. Enveloped VLPs are relatively complex structures consisting of host cell membranes (as envelop) with target antigens on the outer surface. These types of particles provide a higher degree of flexibility for integrating most antigens from the same or different pathogens. The most prominent examples of enveloped viral-like particles are those engineered to express vaccine target antigens from influenza viruses, retroviruses, and hepatitis C virus (HCV) [113].

In the case of the liposomes, the most important challenge is the limited cargo capacity and the fast release of the cargo to the environment. However, these types of nanoparticles have several advantages, such as relative easiness of production, long-term physical stability, and high control over surface properties. These nanoparticles, in conjunction with S and R protein-encoding RNAs, have also been used to control the progression of CoVs such as SARS-CoV [114] and SARS-CoV-2 [115].

Two types of methods (mechanical and non-mechanical) are used to make liposomes and nanoliposomes. The mechanical methods include sonication, homogenization, extrusion, and microfluidization. The non-mechanical methods are reverse-phase evaporation, discharge of lipid–detergent micelle combination, freeze drying, solvent injection, thin layer dehydration, and the thermal method. The sonication method, the most common method in the production of liposomes, uses sound energy to create cavities and disperse particles [116]. The cavities created by the sound effect cause the gas bubbles in the liquid to expand and contract. As the waves increase, the bubbles begin to oscillate and eventually burst. As a result, small vesicles form. This method is used in the preparation of monolayer nanoliposomes. This method is also divided into three types of sonication with a probe, sonication in a water bath, and ultrasonication [117].

As far as exosomes are concerned, they have been used to combat the SARS-CoV virus [118]. In the study, 293 T cells were infected using plasmids containing the gene encoding the S protein. The exosomes obtained from these cells were then used as nanoparticle vaccines against the virus. The clinical research on the efficiency of this approach is in its early stages. The advantages of this method include high biocompatibility, but the high cost of production and the complex and long stages of nanoparticle development are disadvantageous. Scientists in the field have previously used this technology to design vaccines against infectious diseases, such as toxoplasmosis, HIV, influenza, etc. [109,119]. Given this evidence, it may make sense to consider using the same technique against the present virus (SARS-CoV-2).

In a recent study, a polyethylenimine nanopolymer was used to transport SARS-CoV-2 S protein antigens to stimulate immune responses. The use of these antigen-carrying nanoparticles induced immune and humoral responses in mice (production of IgG, IgA, g-interferon, and interleukin-2) [120]. Other researchers have used chitosan polymers to transport plasmids carrying the N protein gene. When these nano-carriers were delivered through the intranasal route, the production of IgG and IgA in mice increased [119]. In an interesting study by Kato et al., a nano-virus particle containing the N, S, and E proteins of MERS-CoV virus was artificially produced. Although no in vivo studies have been performed [109], the study was very important, since it demonstrated that one can fabricate artificial viral particles with the help of nanoparticles.

The most important challenge in using nanoparticle-based vaccines is their cytotoxicity and the need for adjuvants to increase the effectiveness of these vaccines [121]. In another study, using the RBD of MERS-CoV virus and ferritin nanoparticle aggregates, a 28–30 nm nano-scaffold was developed that could immunize mice against the SARS-CoV-2 for a limited time [122]. By and large, ref. [97] is useful for studying different types of nanoparticles in other aspects of dealing with viruses, such as use in hygiene and the prevention of virus pandemics.

### 3.5. Delivery Role of Nanoparticles: Focusing on the CoVs

Relying on human knowledge, which came from the use of DNA and RNA vaccines, the idea of using nanoparticles for vaccination was introduced in the field of COVID-19 treatment. The use of this strategy to deliver small interfering RNA (siRNA) vaccines has helped control the symptoms of several diseases, such as autoimmune and neurological diseases [123]. The fundamentals of the preparation of mRNA-1273 vaccine have been based on this idea. The vaccine comprises an RNA genome covered with a lipid-based nanoparticle envelop [124]. Nanoparticles can also be useful in the delivery of therapeutic antigens. Thanks to the development of science in this field, nanoparticles can be classified into two major types by considering whether the antigen is located inside (encapsulated antigen) or on the surface (surface-presented antigen) of the nanocarrier particle (Figure 5).

Different nanoparticles with different sources can be made depending on whether the therapeutic agent is to be placed on its surface or loaded into it. Nanoparticles containing the DNA genome can penetrate the host cell membrane and enter the cell and eventually arrive at the nucleus. The exogenous gene can enter transcriptional and eventually translational cycles to produce protein products that aid in the healing process. On the other hand, nanoparticles that carry antigens inside them can also enter antigen-presenting cells. After antigen processing by the cell, the antigen eventually would be present on the surface of these cells. These efficient antigens are ultimately able to invoke immune cells and activate immune responses. Alternatively, nanoparticles containing the RNA genome can enter antigen-presenting cells, enter directly into translocation cycles of the cell, and synthesize corresponded antigens. The disadvantage of such nanoparticles is their short half-life, but at the same time, they can produce relatively faster responses compared to DNA-containing nanoparticles. Another group of therapeutic nanomaterials is nanoparticles that carry antigens directly on their surface. These systems interact directly with immune cells. All of these nanoparticles will finally trigger the production of antibodies by immune cells.

RNA- and DNA-containing nanoparticles are generally placed inside nanoparticles due to the degradability of the nucleotides, but antigens can also be placed inside the nanoparticles or on the surface of these carriers [125,126]. The hope for the success of the genome-containing nanoparticles is very high. Among the group of nanoparticles that are being studied to oppose the CoV-2 are Moderna [127], Arcturus Therap [128], and CanSino [14]. There are also many cases of nanoparticles presenting a viral antigen on their surface [125]. The basis of the antigen placement at the nanoparticle surface is to mimic a viral particle. For example, by placing important antigens, such as the S protein, on the surface of a nanoparticle and injecting it into the body, an immune response can be activated against this particle without worrying about the generation of the new viral particles in the body.

### 3.6. Attacking the CoV-2 Life-Cycle with the Help of Nanoparticles

Luckily for us, various steps must be taken to finally create a new virus particle inside a host cell. As illustrated in Figure 6, which shows all the steps of the CoV-2 life cycle, scientists have been able to design various strategies to combat the virus by specifically attacking it at different stages (up to stage 5). To that end, according to the results obtained so far, it is clear that most success in the struggle between the therapeutic agents and the virus can be achieved before the extremely important viral enzyme, RNA-dependent RNA polymerase, starts to function. If the virus cycle cannot be controlled up to this stage, due to the speed of enzyme action and its specific activity and high turnover number of an enzyme, a significant number of viral RNAs will be formed in a short time, and the next steps will proceed at a worrying rate. After this stage, there is almost no hope for blocking the virus life cycle.

It makes sense that all efforts should be made to prevent the virus from entering human cells, but sometimes this is not the only recommended way. Up to now, two drugs have been suggested to inhibit this stage: Umifenovir and Camostat mesylate. The first drug, which is publicly licensed in China (People’s Republic of China) and Russia, can prevent the virus from entering the body by binding to S proteins on the surface of the influenza viruses. In this regard, the drug has also been used as a candidate to prevent the entry of CoV-2 [129]. Camostat mesylate, with a similar mechanism, can prevent the entry of CoV-2 into the host cell [130]. Similarly, in 1994, a direct relationship between inhibition of the furin enzyme and attenuating penetration of influenza viruses to the host cell was proposed [131]. Several furin cleavage sites have been found in the S protein of SARS-CoV-2, which supports the theory of furin-dependent viral entry in the virus [132].

Given the above information, placing this protein on the surface of nanoparticles could serve as a good model for the viral binding inhibition studies. Of the parts of the virus life cycle, the entry of the viral genome into the host cell, led by S2 subunit, is an important stage [133]. As the virus nears the host cell membrane, two heptad repeat regions in S2 (HR1 and HR2) undergo structural changes to facilitate the integration of the virus and host cell membranes [134,135]. So far, many peptides have been made from these two protein regions that can mimic the behavior of these peptides and ultimately prevent the binding of the viruses to the host cells [136,137,138]. As an example, the synthetic peptide HR2 was injected into mice 5 h before the animals were infected with MERS-CoV, and it was observed that the viral infection in the lungs was significantly reduced as a result of such treatment [139].

However, the physical and chemical stabilities of a peptide (especially exogenous peptides) inside a cell are low [140]. Therefore, putting such therapeutic peptides in/on the nanocarrier is expected to provide more promising results. Alternatively, viral proteases can be considered good targets for attenuating the progression of viruses. It has been indicated that the *nsp3* and *nsp5* genes of CoVs encode papain-like cysteine protease (PLpro) and 3C-like serine protease (3CLpro), respectively [141]. In general, the function of these proteases plays a very important role in the process of viral genome transcription and virus duplication [142,143]. Therefore, targeting these proteases could be a key strategy to inhibit CoV-2 infection.

A similar strategy has been used to combat HIV-1. Lopinavir (LPV) and ritonavir (RTV) are two important drugs that can target HIV-1 proteases. Fortunately, these drugs have been shown to target 3CLpro in both SARS-CoV-1 and MERS-CoV viruses and reduce the activity of these proteases [144,145]. The effect of these two important drugs in patients infected with SARS-CoV virus reduced viral load and viral-mediated mortality rate [146]. Also, the combined use of these two drugs in marmoset animal models has been shown to diminish viral loading and improve the general state of the body after infection [147]. However, due to the limited age of the tested animals and the different effects that the SARS-CoV virus had at various ages, the effects of these drugs could not be considered to be the same for all patients [148]. By looking at nanocarriers, their remarkable ability to cross membranes and high-performance delivery of such protease inhibitor drugs could be a promising strategy to overcome viral progression inside a host cell.

Unfortunately, not many mechanisms have been proposed to explain the inhibitory functions of nanoparticles at the cellular level, which indicates a general lack of knowledge and awareness in this area. In general, when talking about the virus outside the cell, it is possible to explore novel mechanisms by utilizing various approaches and designing new experiments. However, research opportunities are rather limited after the nanoparticles enter the cell space, and as a result, a rather restricted set of mechanisms have been explored in the many previously published studies. In the first step of the defense against SARS-CoV-2 virus, nanoparticles can create a protective “barrier” by blocking the entry of viruses utilizing designed functional groups and antibodies against viral antigens to interact with viruses outside the cell space; i.e., before their attachment to the host cell. Such nanoparticles, that are mostly nontoxic toward the host cells, disrupt the SARS-CoV-2 attachment of its receptor and thereby block the process of cell [149]. Another way to achieve this goal is by reducing the expression levels of coronavirus receptors (ACE2 and TMPRSS2) at the cell surface. Unfortunately, no cases were reported, where nanoparticles would be used to down-regulate these receptors to reduce the risk of host cell infection. Also, according to another study, nanoparticles themselves may have a destructive effect on the integrity of the viral structure, preventing the virus from entering the host cell, a result that was seen in the case of silver nanoparticles with a diameter of 2 to 15 nm [150]. However, among the metal nanoparticles studied so far, silver and copper nanoparticles are most frequently used in surface coatings to inhibit the spread of the virus in the environment, whereas gold nanoparticles have been used to combat the virus at the intracellular level [151].

One should keep in mind that since the patients usually receive high doses of the virus, the strategy based on using nanoparticles to prevent the virus from the entry to the host cells may not be able to create a very strong barrier for successful fight against viral infection. Once the virus enters the host cell, the first stage of its intracellular life is to extract the necessary information from the viral genome to synthesize the set of viral proteins. This stage could serve as another station, where nanoparticles should be able to control the function of viruses. At this level, nanoparticles coated with the antisense RNAs can be of great help [152]. Another option includes the utilization of lipid nanoparticles (LNP) as a delivery system for various highly effective small interfering RNA (siRNA) capable of targeting highly conserved regions of the SARS-CoV-2 virus [153]. One of the important considerations for this group of nanoparticles is their capability for rapid and efficient entrance into the host cells. It should also be borne in mind that the toxicity of these particles should be carefully checked to ensure that they do not cause problems entering the cell and coexisting with it for a certain period of time.

One of the attractive targets to attack the SARS-CoV-2 replication by many drugs and nanoparticles is the RNA-dependent RNA polymerase, which plays a central role in the viral infection cycle. At first glance, the fact that this target is an enzyme is very important. The enzyme with high catalytic activity is able to convert a large number of substrates into a product in a very short time (µs range, which is not precisely achieved for SARS-CoV-2 virus). In the case of SARS-CoV-2, RNA-dependent RNA polymerase catalyzes the synthesis of the viral genome from the triphosphate substrates of the host cell. Therefore, nanoparticle can be used to deliver drugs specifically affecting enzymatic activity of this important protein. One should also keep in mind that if viral polymerase cannot be e, the virus will be very difficult to control due to the high rate of synthesis of viral components [154].

The next step that can be affected and potentially controlled by nanoparticles is the efficiency of the viral protein expression. This step is usually conducted by the shared translation machineries of the host and the virus. Therefore, it can be modulated with less efficiency, and corresponding drugs could have more side effects. The final step at which nanoparticles can be expected to help is to prevent viral components from coming together and forming a new virus (packing and budding levels). However, modulation of this stage is not expected to be successful due to the involvement of cellular organelles, such as the Golgi and the smooth endoplasmic reticulum. This could be the reason of why no articles have been published on this subject yet.

It is clear that various side effects of nanoparticles should be considered and carefully analyzed. For example, gold nanorods have been shown to inhibit the mitochondrial degradation of host cells by inhibiting caspase as soon as they enter a host cell infected with SARS-CoV-2 virus, and this intervention ultimately preserves the longevity of the infected cell [71]. A recent study showed that using siRNA/VIPER polyplexes it is possible to slow down the virus replication rate. The study showed promising results at the in vitro (air–liquid–interface) and ex vivo (human lung explant model) levels, and, most importantly, in vivo (lung epithelial cells). It was established that these siRNA/VIPER polyplexes not only were mostly non-toxic to the host cells, but were able to noticeably reduce viral replication [155]. Also, some studies, have found a close link between the pathogenicity induced by different viruses and the rate of autophagy in the host cells of infected people. Therefore, another way to fight the coronavirus is to use nanoparticles functionalized with autophagy inhibitors. Among these drugs, chloroquine and hydroxychloroquine have been able to inhibit autophagy by interfering with the connection of autophagosome to lysosome. The use of these drugs against viruses such as HIV, SARS-CoV, and Zika is also recommended [156].

## 4. Nano-Based Diagnostic Tests for COVID-19

Although a large fraction of the applications of nanoparticles has been in the field of drug delivery and development of SARS-CoV-2 vaccines, some researchers have also focused on the use of nanoparticles for diagnostic and treatment purposes. In the line of the diagnostic applications, the goals can be divided into two major categories: the use of biomolecular corona and colorimetric methods for corona detection. In the first case, research and efforts are hypothetical. Here, with the entry of nanoparticles into the complex and crowded living system (cell), a series of biomolecules such as proteins, polysaccharides/carbohydrates, and lipids will be in contact with the surface of nanoparticles and will form different patterns of bonding at different health conditions. Nanoparticles facing different proteins and biomolecular contents, such as those seen in patients infected with SARS-CoV-2, can form diverse microarrays of biomolecules (mainly proteins and peptides) in different states of health, and based on the analysis of the corresponding results, people with high risk to virus infection can be found [157]. Although numerous articles have already been published in the field of application of this approach in various diseases, no such research has been conducted yet for the SARS-CoV-2-related cases.

The second category of diagnostic research involves colorimetric (visible and/or non-visible) experiments with nanoparticles for the rapid and accurate detection of coronavirus in laboratory samples. In line with these considerations, gold nanoparticles are repeatedly used in the SARS-CoV-2 diagnosis, both in solution and in the form of nanochips. In a study using gold nanoparticles containing SARS-CoV-2 RNA antisense strands, a colorimetric system was developed for SARS-CoV-2 diagnostics without the need for complex instruments [158]. In this study, based on the interaction between suspended gold nanoparticles solution and viral RNA, after a period of incubation, the nano-system turns into larger aggregates, and finally, with the change in resonance of the surface of aggregated gold nanoparticles, diagnosis would be achievable [158]. In general, the physicochemical properties of nanoparticles enable them to be a tool that can produce a measurable signal once they interact with the virus (the interaction between the surface nanoparticles and part of the coronavirus such as surface proteins, viral DNA, and/or RNA). In another study that also used gold nanoparticles, the nanoparticle surface was functionalized using α, N-acetyl neuraminic acid, a specific sugar for binding to the SARS-CoV-2 S protein [159]. In this study, although virus-like particles containing surface S protein were used instead of the whole virus, and the results were promising. In detail, after the nanoparticle and virus complex is formed, it flows in the lateral flow immunoassay strip containing α,N-acetylneuraminic acid sugars. Followed by the formation of clots due to S protein re-interaction with the immobilized sugar on the immunoassay pathway, a signal to detect virus moiety will be produced. From the statistical viewpoint, it is stated that the detection sensitivity of this technique is high, and as little as 5 micrograms of S protein per milliliter can be detected [159].

Although more attention has been paid to the gold nanoparticles for the detection of SARS-CoV-2, some studies have looked at other groups of nanosystems as well. In one such study [160], poly (lactic-co-glycolic) (PLGA) nanoparticles conjugated to viral S protein were used. This system, which is very similar to the immunoassay sandwich, begins with the immobilization of antibodies against S protein in microplates. Then, S protein conjugated nanoparticles are incubated with the antibody, and finally, with the help of the peroxidation-like activity of copper nanoparticles and the external presence of hydrogen peroxide, 3.3%, 5,5′-tetramethylbenzidine immobilized in PLGA oxidases to change the color of the solution. This system is said to detect the S protein at the concentration of a femtogram per milliliter [160].

As aforementioned, the RNA content of SARS-CoV-2 has also been used for diagnostic purposes. In one of the related studies, europium-chelate nanoparticles (FNPs) conjugated to S9.6 antibodies (S9.6-FNPs) were used to detect the presence of SARS-CoV-2 using its RNA. S9.6 antibody is a protein that can bind to DNA and RNA hybrids. In this study, first, a DNA probe that could bind specifically to the viral RNA was used, and then S9.6-FNPs nanoparticles were used to bind to this hybrid. Based on the results of the analysis of samples taken from 734 patients, it was concluded that this test with its 99% specificity and 100% sensitivity could be a good option for the virus diagnosis [161]. Not only viral components but also host molecules released due to the viral entry into the body have been used to diagnose SARS-CoV-2 infection. In fact, diagnosis of immunoglobulin M (IgM) and immunoglobulin G (IgG) has been utilized as another diagnostic option using conjugated nanoparticles with anti-immunoglobulins [162,163].

Looking at the articles published so far, the strategies used in the diagnosis of SARS-CoV-2 by nanoparticles and/or fluorescently labeled nanoparticles can be summarized as shown in Figure 7 [164].

Although nanoparticles were successfully used for the rapid and accurate detection of SARS-CoV-2 in many studies, one of the challenges of these studies has been the high cost of the consumables required for conducting them. Most studies have used gold nanoparticles or nanoparticles labeled with fluorophores, which ultimately increases the cost of each test for people in the community. There is currently no method that is accurate, fast and cheap, but in this challenging era, the existing options are considered as the only available solutions.

Compared to the conventional SARS-CoV-2 detection methods, such as RT-PCR, nanoparticles can be more appropriate options in several respects. For example, by amino functionalizing (via 3-aminopropyl) the surfaces of magnetic nanoparticles, the RNA contents of SARS-CoV-2 can be co-precipitated in a rapid and efficient manner [165]. Likewise, quantum dots have proven their usefulness in many medical applications [166]. These particles are respectable options for studying the binding dynamics of the S protein to the cell surface receptor (ACE2). These particles can be easily detected as soon as they enter the host cell due to their very small size (between 1 to 10 nanometer) and photo-stability [167]. Quantum dots can also be used as probes in the study of inhibitors of binding between the S protein and its receptor [168]. Graphene, which has also been used in many diagnostic aspects [169], was studied in the field of SARS-CoV-2 diagnosis as well. In a corresponding study, S protein antibodies were attached to the surface of graphene using 1-pyrenebutyric acid N-hydroxysuccinimide ester linkers. The system was then used as a sensor to detect a protein with an accuracy of 1 femtogram of the S protein per milliliter. The diagnostic method was based on the field effect transistor (FET) [170].

Furthermore, different nanoparticles were also used in other platforms to fight the SARS-CoV-2. Some nanoparticles have been used as drug carriers against the virus, such as polymeric lipid nanoparticles and silica nanoparticles [171,172,173]. Silver nanoparticles [150] and nanoparticles capable of generating free radicals [174] have also been used in those studies for the killing the virus directly. In other experiments, specific nanoparticles, such as nano-exosomes having the viral contents, were used to neutralize the virus infection in the body [175].

## 5. Conclusions and Future Outlook

It would be very frightening to imagine the epidemic of another virus with SARS-CoV-2-like infectivity and Ebola-like mortality (around 90%) [176]. After taking this alarm seriously, many researchers put more focus on the investigation of various SARS-CoV-2 outbreak issues. In general, to combat any dangerous threats, the most important factors must first be carefully identified and then characterized. By the same token, to deal with the CoVs, exact structural knowledge of viral components and the molecular mechanisms of virus infection and stages of the life cycle must be understood. Some treatment strategies are purely preventative (prophylactic strategies), while others fight infection at different stages of the virus life cycle. Antigens and the viral genome can be used as functional vaccines [15,177,178,179]. While experimental observations from the use of vaccines have been promising, these therapeutic agents also have some challenges, such as short half-lives and unwanted immunological symptoms [124,148]. Recent studies indicate that the utilization of nanoparticles would be also able to help humans. The use of these new materials will enable researchers to meet the challenges of vaccines and raise hopes for the fight against the CoVs. Despite all the positive results and observations that have been made about the use of nanoparticles in the production of vaccines, due to the rush to make the SARS-CoV-2 vaccines, the toxicity of nanoparticle-based vaccines, their targeting accuracy, distribution in the body, and localization within the cell has not been properly assessed. Considering the need for a global vaccination and the fact that only some countries are able to produce vaccines, one of the problems that is expected to be solved in the future is the transfer of vaccine production technology to other countries or the investment of current companies in other companies to upgrade them to produce vaccines. In the process of controlling patients, such as COVID-19, vaccination will be effective when all sections of the population are vaccinated quickly for the sake of reducing the rates of the virus spread and mutation. On the other hand, looking at the origin of this virus, which is still in a state of ambiguity, the guess that it was transmitted from animal to human is not far from doubt. Therefore, the points of view on the lifestyle in the modern world should be re-examined to better control interactions between humans and animals so that in the future, we do not see the transmission of another disease from animals. More detailed studies on human food patterns should also be conducted. Last but not least, the training necessary to improve the scientific level of people in the community to accept vaccines should be increased, so that the doubts about vaccines are reduced, and those who are afraid of vaccination know the danger of this way of thinking for themselves and others.

## Figures and Tables

**Figure 1 nanomaterials-12-01602-f001:**
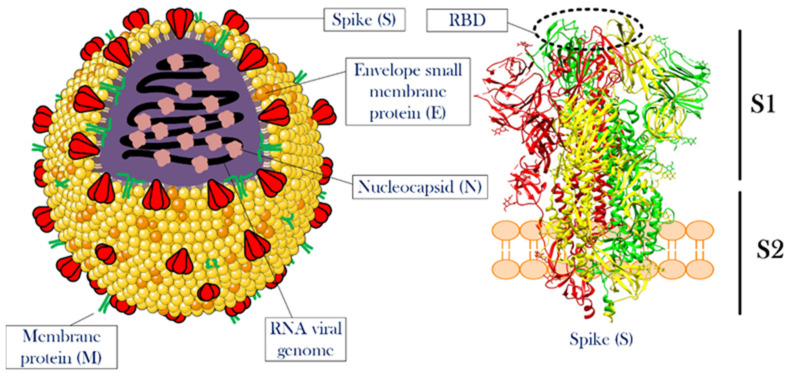
**The overall structure of CoV and the spike protein.** The left side illustrates a single particle of CoV. Three membrane proteins and RNA viral genome in a complex with nucleocapsid proteins were shown in the scheme. The right side shows the detailed structure of a spike protein. This protein, which is the most important functional protein during the attachment of the viral particle to the host cell, has two subunits named S1 and S2. The receptor-binding domain is located at the top of S1. For more information see the text. The structure of the spike protein was extracted from PDB 6ZGI [29]. RBD is receptor-binding domain.

**Figure 2 nanomaterials-12-01602-f002:**
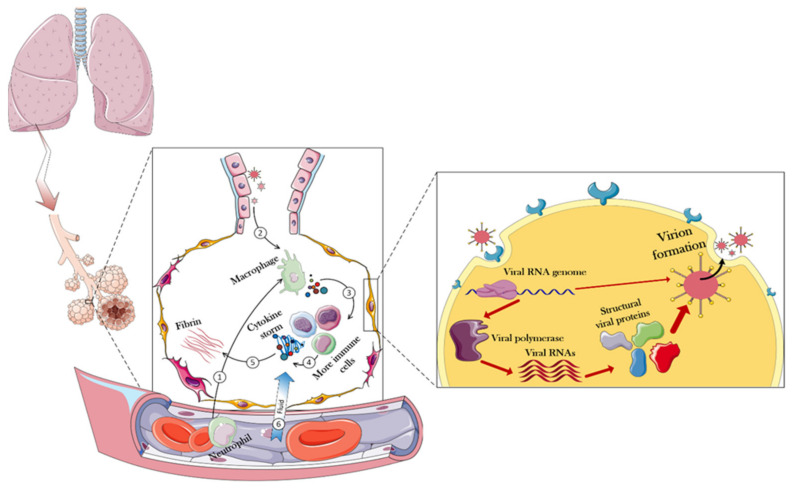
**Immune responses and overall intracellular events triggered in the face of CoV-2.** In normal conditions, immune cells enter the alveoli of the lungs through the blood. When CoV arrives in the alveolar compartment, these immune cells oppose them and eventually produce cytokines. Positively regulating, cytokines trigger more immune cells, eventually producing more cytokines. During these events, the fibrin of the alveoli increases, resulting in partial destruction and increased permeability of the alveoli. Consequently, fluid goes on the battlefield (alveoli) through the capillaries, causing destruction and edema of the infected lung. However, when a virus can compete with these challenges, it will be able to enter the host cell. In the host cell, the viral genome is generally released and the host equipment is used to replicate the viral particles. The following sections of the article provide more details in this regard.

**Figure 3 nanomaterials-12-01602-f003:**
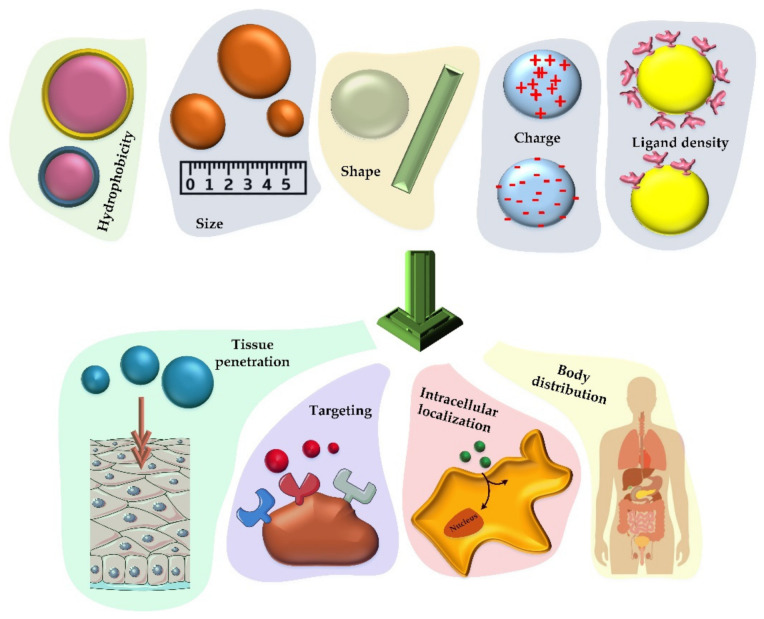
**Various chemical and physical properties that can determine how nanoparticles will act as vaccine carriers.** The size, hydrophobicity, charge, shape, and ligand density of the nanoparticles that carry the therapeutic factor will have a significant effect on their cellular uptake, distribution in the body, accumulation in the cell (especially phagocytic cells) and the rate of tissue penetration.

**Figure 4 nanomaterials-12-01602-f004:**
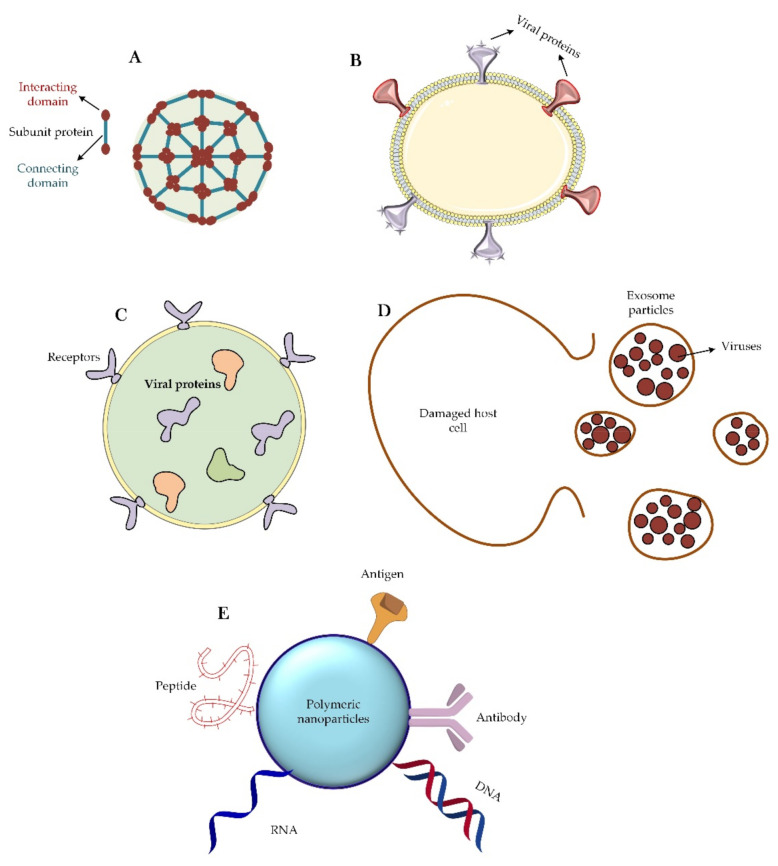
**Different classes of nanoparticles are used as virus vaccines.** (**A**) Self-assembling capsid protein nanoparticle. This type of nanoparticle is made up entirely of proteins that are able to self-aggregate. Sometimes two or three types of proteins are used to make this nanoparticle. (**B**) Virus-like particle. Particle engineering has created the ability to design and synthesize a virus-like particle that is an assembly of a phospholipid and a set of viral proteins. (**C**) Liposome. Liposomes are free of any viral proteins on their surface. Sometimes they may have receptors for the correct targeting of the particle, but it should be noted that viral proteins are trapped inside the liposome structure and enter the immunological pathways into the host cell after the endocytosis of the particle. (**D**) Exosome particle. Once the host cell is infected with the virus, an exosome will emerge from the damaged cell that contain the newly synthesized viruses. These particles, after extraction and purification, can be suitable treatment options. (**E**) Corresponds many available nano-based polymeric materials that functionalized with different therapeutic agents such as DNA, RNA, antigens, peptides, and antibodies.

**Figure 5 nanomaterials-12-01602-f005:**
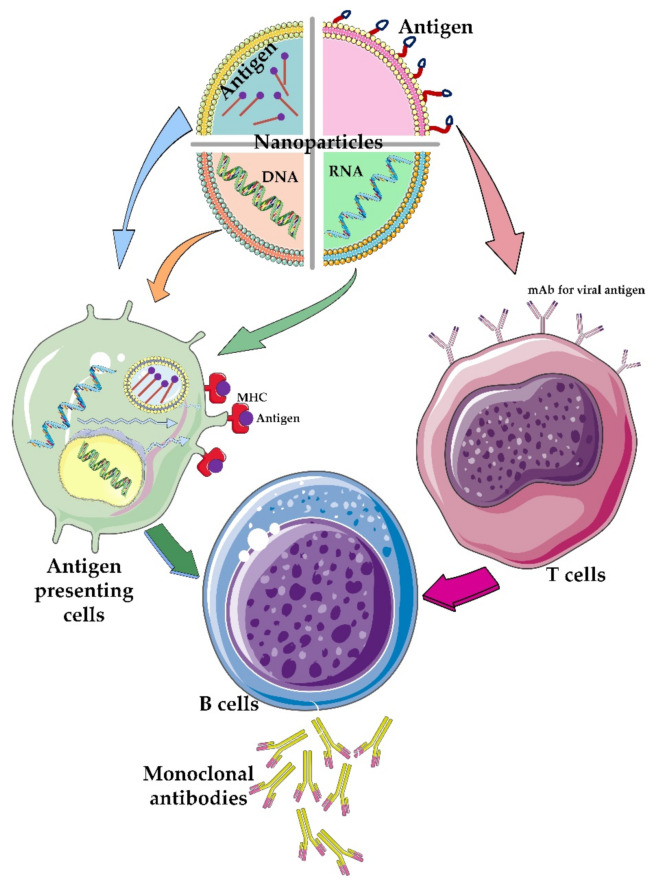
**The role of different nanoparticles containing therapeutic agents.** Different nanoparticles have been used to carry different components of viral particles such as genetic material and/or its antigens. The viral genetic materials are usually encapsulated or trapped inside the nanoparticles while the viral antigens are functionalized on the surface of the nanoparticles. Depending on which type of T-cell or antigen-presenting cells these engineered nanoparticles attach to, different immunological pathways are created in the body, which ultimately lead to the activation of B cells that produce monoclonal antibodies against the virus particle.

**Figure 6 nanomaterials-12-01602-f006:**
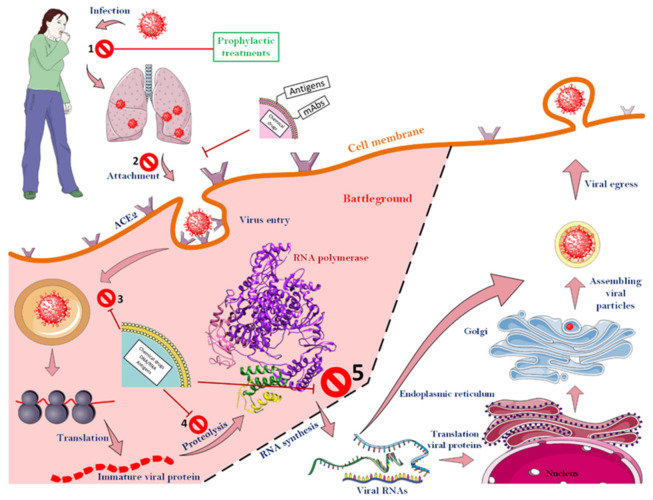
**Different stages of a CoV-2 life cycle along with target points for fighting the virus.** The CoVs can be fought almost from the beginning of the virus to the function of the RNA-dependent RNA polymerase. However, after the function of the enzyme, no specific strategy has been proposed to deal with the virus. As soon as the virus enters the body, a range of events will occur, though up to stage, 5 the virus can still be defeated. Nanoparticles that have monoclonal antibodies or antigens on their surface usually act before the virus enters the cell. The ACE2 receptor, which is among the most important cell surface receptors for the SARS-CoV-2, was a target of many studies for blocking virus cell entry. As soon as the virus binds to this receptor, the process of virus penetration into the host cell begins. Consequently, masking this agent on the surface of host cells can prevent the virus from entering the cell. Nanoparticles that encapsulated therapeutic agents will be able to fight the virus as it enters the host cell. See the text for more details.

**Figure 7 nanomaterials-12-01602-f007:**
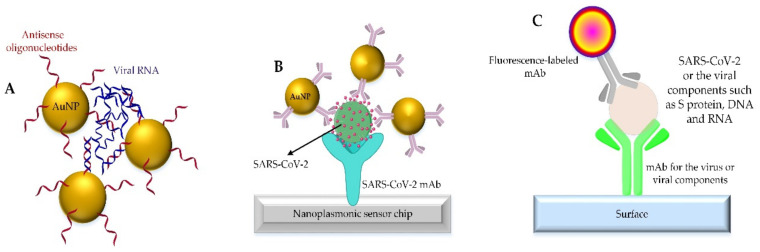
Different strategies used for the detection of SARS-CoV-2 by nanoparticles (**A**,**B**) and/or fluorescently labeled nanoparticles (**C**).

**Table 1 nanomaterials-12-01602-t001:** Previously used different classes of nanoparticles in respiratory viral diseases.

Compound	Virus	Antigen	Adjuvant	NP Size (Diameter, nm)	Outcome
Polyanhydride	RSV	G and F glycoproteins	-	200–800	The replication of virus was suppressed in infected mice
HPMA/NIPAM	RSV	F protein	TLR-7/8 agonist	12–25	By having significant antigenicity, TH1 isotype anti-RSV F antibodies was produce in the blood.
Chitosan	IF(H1N1)	IF(H1N1)	Heat shock proteins	200–250	After administration, the nanosystem produced antibody and induced T cell immunity.
PLGA	BPI3V	BPI3V proteins	-	225.4	The infected pigs had low virus penetration (loading) in their lungs.
Gold	IF	Antigen M2e	CpG	12	Full protection of vaccinated mice against the virus by the increasing M2e-specific IgG in serum.
Q11 peptide	IF(H1N1)	Antigen M2e	-	15–100	Protection against homologous challenge of IF PR8 H1N1 and heterologous challenge of avian IF H7N9.
Viral-like particle	RSV	M1 protein of IF and RSV-F or -G	MPL and trehalose 6,6 dimycolate	10–1000	Induction the memory of T cell responses.

Abbreviations: RSV, respiratory syncytial virus; TLR, toll-like receptor; TH1, T helper type 1; PLGA, Poly(d,l-lactide-*co*-glycolide).

## Data Availability

Not applicable.

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
