# Peer review of "Nanoparticles for Coronavirus Control"

_nanomaterials, 2022, doi:10.3390/nano12091602_

Round 1
Reviewer 1 Report
In this review paper, the authors described the coronavirus causing COVID-19 (SARS-CoV-2) and vaccines from a nanomaerial aspect. Nano-based diagnostic is also described. It is an interesting review since it focuses both on disease and vaccine from the same aspect. I would recommend publication after the following issues are addressed.
1) Figure 3 should be revised. Fig. 3A looks virus-like particles and Fig. B looks cells. Figure C and D look similar with additional unknown components in D. Four types of nanoparticles should be illustrated in a comparative but distinguishable manner. Each component such as proteins should be shown similarly.
2) Various nanoparticles described in Page 7 (line 247-263) should be shown as figures (maybe in Fig.3 as recent examples ?).
3) Figures corresponding to the section 4 (Nano-based diagnostic tests for COVID-19) should be added.
(minor issues)
4) Each panel (A-D) in Figure 3 should be quoted in the main text.
5) ACE2 receptor should be described in the figure(s) since it is the key component.
6) Page 7 line 274 Figure 6 -> Figure 4 (?)
7) Explanation should be added in the Figure 4 legend.
Author Response
With eternal appreciation for your efforts, please find the attachment, the response to reviewers. Since some comments of reviewers were along the same lines, the other comments were also attached here as a single file.

Reviewer 2 Report
The authors mentioned the details regarding the structure and vaccine developments clearly. But I would like to get to your reference that, the data you suggest is not up - to -date and I request the authors to see the recent publications and also clinical data regarding vaccines and their developments.
Comment 1 The details of the information regarding the peptide-based nanoparticles, Polymeric based nanoparticles, and Inorganic and metal nanoparticles are to be discussed.
Comment 2 A tabular column is needed for nanoparticle based tools used against Corona Virus, types of virus adjuvant, and applications with a detailed description
Comment 3
As the title is given Nanoparticles for Corona Virus Control, the authors mentioned only the structure of the Corona Virus. But they did not mention the substantial involvement and the role of nanoparticles with Corona Virus.
Comment 4 The conclusion part mentioned here was only regarding the data that was collected in the past. It would be suggested to get your reference that there was a total of 60-80% nanoparticle-based vaccines undergoing clinical trials and approvals? Do the authors have to justify?
It would be good if the authors modifies the articles according to the above comments and also include the current and future prospectus
Author Response
With eternal appreciation for your efforts to improve the manuscript, please refer to the attached file. Since some of the reviewer's comments were in line with others, all the answers were provided in one file.

Reviewer 3 Report
The MS entitled "Nanoparticles for Coronavirus control" is interesting,however, it needs discussion in the MS to be accepted to this journal
which was submitted being major corrections as following.
- The article should include a discussion about the advantages of using nanoparticles as a vaccine against a virus and specifically against SARS-CoV-2 than other types of vaccines and detection techniques.
- In this sentence "For this important purpose, a variety of nanoparticles, such as carbon-based nanoparticles, nanopolymers, liposomes and quantum dots have been used [67]" This statement lacks references to each type of nanomaterial .
- Explain the different methods of preparation of different nanoparticles and how viable it would be prepared as a vaccine in the world ??? and also as a detection test.
4. Add statistics on how effective a nanoparticle is as a vaccine and as a virus detection in humans based on previous research
Author Response

(The authors gave the same response as above.)

Reviewer 4 Report
The paper is good and no need to review at all I think!
Author Response

(The authors gave the same response as above.)

Reviewer 5 Report
The manuscript ID nanomaterials-1670830 mainly presents a review related to particular descriptions of the SARS-CoV-2 and biotechnological methods of vaccine production involving nanoparticles. Please see below a list of points to the authors:
- In the introduction must be reported what nanoparticles were selected to be analyzed in this review.
- The advantages and disadvantages of nanoparticle-based vaccines for CoVs must be described in detail.
- Do the nanoparticles must be functionalized for these delivery and attacking applications? Please describe and justify.
- Please report the size of the different nanoparticles selected to be part in this review.
- It would be helpful to analyze how should be the selection of the shape and size of the nanoparticles in the topic of biotechnological methods of vaccine production.
- Including perspectives related to nano-based diagnostic tests for COVID-19 before the conclusions would improve the presentation of the work.
- The title states “coronavirus control.” The authors are invited to discuss about some mechanisms with potential interplay during nanoparticles action in this topic. You can see for instance doi:10.3390/cells9122679
- The collective citations should be presented in single form in order to clearly justify them by individual expressions associated with this topic.
- The report should clearly state what this review adds in respect to comparative reviews made in topics related to SARS-CoV-2 considering the nanomaterials science field, or be of relevance to nanotechnology. You can see for instance https://doi.org/10.1016/j.cossms.2021.100964
- The list of references must be updated to 2022.
Author Response

(The authors gave the same response as above.)

Round 2
Reviewer 5 Report
I appreciate the effort of the authors to address the points raised in the review stage. However, in my opinion, a fundamental point requested previously as number 7 in my review is still missing. Please see below this issue rewritten for the authors:
*The title states “coronavirus control.” But no discussion about the nanoscale mechanisms related to coronavirus control is present in the report. The potential interplay of coronavirus during nanoparticles action in this topic should be deeply analyzed.
Author Response
RESPONSE: Thank you for pointing this out. The requested data are included in lines 578-649.